

# Quantifying fugitive gas emissions from an oil sands tailings pond with open-path FTIR measurements

Yuan You[1,§], Samar G. Moussa[1], Lucas Zhang[2], Long Fu[2], James Beck[3], Ralf M. Staebler[1]

[1] Air Quality Research Division, Environment and Climate Change Canada (ECCC), Toronto, M3H 5T4, Canada
[2] Alberta Environment and Parks, Edmonton, T5J, 5C6, Canada
[3] Suncor Energy Inc. , Calgary, T2P 3Y7, Canada
[§] Now at Department of Physics, University of Toronto, Toronto, M5S 1A7, Canada

*Correspondence to:* Ralf M. Staebler (ralf.staebler@canada.ca)

**Abstract.** Fugitive emissions from tailings ponds contribute significantly to facility emissions in the Alberta Oil Sands, but details on chemical emission profiles and the temporal and spatial variability of emissions to the atmosphere are sparse, since flux measurement techniques applied for compliance monitoring have their limitations. In this study, open-path Fourier transform infrared spectroscopy was evaluated as a potential alternative method for quantifying spatially representative fluxes for various pollutants (methane, ammonia, and alkanes) from a particular pond, using vertical flux gradient and inverse dispersion methods. Gradient fluxes of methane averaged 3.7 g m$^{-2}$d$^{-1}$ but were 40% lower than nearby eddy covariance measurements, while inverse dispersion fluxes agreed to within 11%. Significant NH$_3$ emission fluxes were observed (0.11 g m$^{-2}$d$^{-1}$ (92 tonnes y$^{-1}$)), and total alkane fluxes were estimated to be 1.33 g m$^{-2}$d$^{-1}$ (1120 tonnes y$^{-1}$), representing 12% of the facility emissions.

## 1 Introduction

Tailings from the oil sands industrial processes in Alberta's Athabasca Oil Sands consist of a mixture of water, sand, non-recovered bitumen, and additives from the bitumen extraction processes (Small et al., 2015). These tailings are deposited into large engineered tailings ponds on site. Separation of processed water from remaining tailings occurs continuously in the tailings pond, and the processed water is recycled (Canada's Oil sands. Tailings Ponds: https://www.canadasoilsands.ca/en/explore-topics/tailings-ponds). The total liquid surface area covered by tailings ponds in the Athabasca Oil Sands was 103 km$^2$ in 2016 and continues to grow (Alberta Environment and Parks, 2016). Emissions to the atmosphere from tailings ponds include methane (CH$_4$), carbon dioxide (CO$_2$), reduced sulfur compounds, volatile organic compounds (VOCs), and polycyclic aromatic hydrocarbons (PAHs) (Siddique et al., 2007; Simpson et al., 2010; Yeh et al., 2010; Siddique et al., 2011; Siddique et al., 2012; Galarneau et al., 2014; Small et al., 2015; Bari and Kindzierski, 2018; Zhang et al., 2019). Emissions from tailings ponds vary with pond conditions, such as pond age and solvents additives in the ponds, and can contribute significantly to total facility emissions (Small et al., 2015).

Very few studies focusing on emissions of air pollutants from tailings ponds have been published (Galarneau et al., 2014; Small et al., 2015; Zhang et al., 2019). Compounds of particular interest include alkanes and ammonia (NH$_3$). Alkanes are part of the solvents used in the extraction process (Small et al., 2015), and can dominate VOCs emissions



from oil sands facilities (Li et al., 2017). Previously reported VOCs emissions by facilities had large uncertainties, especially from fugitive sources, due to limitations of the methods used to estimate emissions for compliance monitoring purposes (Li et al., 2017). VOCs in the atmosphere are important because of their effects on ambient ozone and secondary aerosol formation (Field et al., 2015; Kroll and Seinfeld, 2008). Emissions of $NH_3$ from tailings ponds

to the atmosphere have not been published, although $NH_3$ has been observed in the oil sands region (Bytnerowicz, et al., 2012; Whaley et al., 2018). $NH_3$ emissions have important environmental implications, such as forming atmospheric aerosols with sulfuric acid (Kürten, et al., 2016) and affecting nitrogen deposition in the ecosystem (Makar, et al., 2018). This information is important for model simulations of critical loads of acidifying deposition in the ecosystem (Makar, et al., 2018). This field measurement project provided a great opportunity to continuously

measure and to quantify tailings pond emissions over more than a month, especially for $NH_3$ and total alkanes.

Open-path Fourier transform infrared (OP-FTIR) spectroscopy has been considered a good candidate for an alternative method to monitor fugitive emissions from industrial or hazardous waste area sources, since the method is non-intrusive, integrates over long path-lengths, and has the ability to quantify several different gases of interest simultaneously and continuously (Marshall et al., 1994), without sample line issues . It has previously been used to

quantify mole fractions of various air pollutants from different sources such as forest fires (Griffith et al., 1991; Yokelson et al., 1996; Yokelson et al., 1997; Goode et al., 1999; Yokelson, 1999; Yokelson et al., 2007; Burling et al., 2010; Johnson et al., 2010; Akagi et al., 2013; Yokelson et al., 2013; Akagi et al., 2014; Paton-Walsh et al., 2014; Smith et al., 2014), volcanoes (Horrocks et al., 1999; Oppenheimer and Kyle, 2008), industrial sites (Wu et al., 1995), harbours (Wiacek et al., 2018), and road vehicles (Bradley et al., 2000; Grutter et al., 2003; You et al., 2017). OP-

FTIR measurements with vertically separated paths have previously been conducted to derive emission rate of air pollutants. Schäfer et al. (2012) deployed two OP-FTIR spectrometers with parallel paths 2.2 meter vertically apart at a grassland at Fuhrberg Germany, to measure nitrous oxide ($N_2O$) emissions with a flux-gradient method, and showed the calculated flux is comparable to the chamber measurements at the same grassland. Flesch et al. (2016) deployed OP-FTIR measurement with one spectrometer and two paths vertically separated by about 1 m on average ("slant

path" configuration) at a cattle field in Alberta, Canada. They derived emission rates of $N_2O$ and $NH_3$ by flux-gradient and inverse-dispersion methods, demonstrating the capability of OP-FTIR systems to measure emission rates of $N_2O$ and $NH_3$. Following the flux-gradient method in Flesch et al. (2016), Bai et al. (2018) measured the flux of $N_2O$, $NH_3$, $CH_4$, and $CO_2$ from a vegetable farm in Australia by an OP-FTIR system with two paths vertically separated by 0.5 m on average. At the same vegetable farm, Bai et al. (2019) measured emission rates of $N_2O$ by flux chambers and OP-

FTIR "slant path" configuration with flux-gradient methods, and showed a large variation of the ratio of $N_2O$ fluxes with these two measurements. Inverse dispersion models have also been applied to OP- FTIR measurements to quantify emission rates in previous studies (Flesch et al., 2004; Flesch et al., 2005; Bai et al., 2014; Hu et al., 2016; Shonkwiler and Ham, 2018).

Longer continuous coverage with a greater height difference between paths is one distinguishing feature of this study

compared to previous research. The motivation of this work is to quantify emission rates of pollutants from one specific tailings pond by combining OP-FTIR measurements with micrometeorological methods. Emissions of $CH_4$, $NH_3$, and total alkanes as well as a comparison of gradient and inverse dispersion methods are presented in this study.



## 2 Open-path FTIR field measurements and methods for deriving fluxes

The main site of this study was on the south shore of Suncor Pond 2/3 (Fig. S1; 56°59'0.90"N, 111°30'30.30"W 305m
ASL). Turbulent fluxes were measured on a mobile tower with sonic anemometers (model CSAT-3, Campbell
Scientific, USA) at three levels at 8m, 18m, and 32m above ground. Vertical gradients of gaseous pollutant mole
fractions were measured by drawing air from 4m, 8m, 18, 32m to instrumentation housed in a trailer on the ground.
The sample inlet at 4m was on the roof of the main trailer beside the mobile tower. Amongst these instruments for
gaseous pollutants, cavity ring-down spectroscopy (model G2311-f and model G2204, Picarro, USA) were included
to measure the $CH_4$ eddy covariance flux and $CH_4$ mole fraction vertical profile and calibrate the mole fraction from
OP-FTIR retrievals. A propeller anemometer (Model 05103-10, Campbell Scientific, USA) on the roof of the main
trailer at 4m above ground provided an additional measurement of wind speed and direction. Measurements were
conducted from July 28 to September 5, 2017. The FTIR spectrometer was located right beside the flux tower and the
paths were along the south shore of the pond. This manuscript focuses on derived fluxes from the measurement of
OP-FTIR. Other experimental details of the project can be found in You et al (2020).

### 2.1 Open-path Fourier transform infrared spectrometer (OP-FITR) system

The FTIR measurements were taken with a commercial Open Path FTIR Spectrometer (Open Path Air Monitoring
System (OPS), Bruker, Germany), which was setup at 1.7m above the ground in a trailer. The infrared source is an
air-cooled Globar. The emitted radiation is directed through the interferometer where it is modulated, travels along
the measurement path (200m horizontal distance) to a retroreflector array that reflects the radiation, travels back to
the spectrometer, and enters a Stirling-cooled mercury cadmium telluride (MCT) detector (monostatic configuration).
Three retroreflectors were employed in this study: one near ground level (1m) on a tripod, and two at higher elevations
on basket lifts, resulting in heights of reflectors of approximately 1m, 11m and 23m above ground. Three paths with
these three retro-reflectors are referred as bottom, middle and top paths. The bottom retro-reflector was approximately
twice the size of the upper two (59 vs. 30 reflector cubes). All retro-reflectors were cleaned with an alcohol solution
once during the study, and the bottom mirror were rinsed with de-ionized water three times. Return signal strength
decreased by around 65% during the 5-week study due to reflector deterioration, presumably mostly due to impaction
by particulate matter. This reflector deterioration also decreased the signal-to-noise by around 67%, based on spectral
retrievals for $CH_4$ , but did not affect the mean mole fractions measured.

In this study, spectra were measured at a resolution of 0.5 cm$^{-1}$ with 250 scans co-added to increase signal-to-noise
ratio, resulting in roughly a one-minute temporal resolution. Stray light spectra were recorded regularly by pointing
the spectrometer away from the retroreflectors. This stray light spectrum accounts for radiation back to the detector
from internal reflections inside the spectrometer, i.e. not from the retroreflector array, and was subtracted from all the
measurement spectra before performing further analysis.

Spectral fitting was performed with OPUS_RS (Bruker), which uses a non-linear curve fitting algorithm (You et al.,
2017). Spectral windows and interference gases for each gas (Table 1) were determined by optimizing capture of the
absorption features while minimizing interferences. To further improve fittings, baselines were optimized through
either linear or Gaussian fits under given spectral windows and interfering gases. For $CH_4$ and $CO_2$, temperature-



dependent reference files were used for fitting and retrieving mole fractions. For other pollutants, reference spectra at
296K were used and retrieved mole fractions were corrected for air density using measured ambient temperature and
pressure. Retrieved $CH_4$ mole fractions from FTIR were then calibrated against $CH_4$ mole fractions from point cavity
ring-down spectrometer (CRDS) measurements (Picarro G2204) at 4 m (Supplemental Material 2.1. Fig. S2). These
calibrated $CH_4$ mole fractions from the FTIR were then used in flux calculations.

We also attempted to retrieve several other pollutants from measured FTIR spectra, but encountered insufficient
signal-to-noise ratios, given the existing mole fractions at this location, variability in ambient $H_2O$ vapor, etc. These
pollutants include toluene, benzene, xylenes, sulfur dioxide, dimethyl sulfide, carbonyl sulfide, formic acid, and
hydrogen cyanide. For these trace gases at this site, the detection limits of this open-path system were insufficient for
flux calculations.

**2.2 Method of deriving gradient flux**

**2.2.1 Gradient flux**

Gradient flux estimates are derived from the vertical gradient of mole fractions and the associated turbulence, given
by

$$F_c = -K_c \frac{\partial c}{\partial z} \tag{1}$$

where $F_c$ is the gradient flux for a pollutant $c$, and $\frac{\partial c}{\partial z}$ is the vertical gradient of mole fractions. $K_c$ is the eddy diffusivity,
a transfer coefficient characterizing turbulent transport (Monin and Obukhov, 1954). In this study, the gradient flux
of pollutants measured by the OP- FTIR system is calculated with the "modified Bowen Ratio" method (Meyers et
al., 1996; Bolinius et al., 2016). Details on the calculation of eddy diffusivities and eddy covariance fluxes of $CH_4$,
can be found in You et al. (2020).

In this study, vertical profiles of the $CH_4$ mole fractions varied over time and mostly showed linear vertical profiles
when the wind was from the pond (Supplemental Material Section 2.2). In the following calculation, the vertical
profiles of $CH_4$ and other gases are considered linear over the entire project. Therefore, the representative average
height of the FTIR top path is taken as the height of the middle point (at 12 m). For a given period, $K_c$ is dependent
on the height above the surface (Monin and Obukhov, 1954). $K_c$ for gradient flux calculated from the top and bottom
FTIR paths has been adjusted linearly based on the $K_{c\_2,4}$ calculated from point measurements at 8m and 32m on the
tower:

$$\frac{K_{c\_FTIR}}{K_{c\_2,4}} = \frac{\frac{1+12}{2}}{\frac{8+32}{2}} = 0.325 \tag{2}$$

The gradient flux is calculated by combining this $K_{c\_FTIR}$ with the mole fractions gradient between top and bottom path
of FTIR in Eq (1). The difference between assuming linear and logarithmic vertical profiles of the mole fractions is


discussed in the Supplemental Material Section 2.2. The logarithmic vertical profile assumption resulted in fluxes that
were on average 85% of the gradient flux calculated with linear vertical profiles.

In addition to calculating gradient fluxes by using $CH_4$ mole fractions gradient between top and bottom paths, gradient
fluxes of $CH_4$ were also calculated by using mole fractions gradient between middle and bottom paths. Results show
that gradient fluxes with top-bottom paths gradient and with middle-bottom paths gradients are consistent within 95%
(Supplemental Material Section 2.2, Table S1). These results suggest the gradient fluxes in this study are not sensitive
to which paths were chosen, and support the assumption of linear profiles.

### 2.2.2   Inverse dispersion fluxes

Inverse dispersion models (IDMs) can be used to derive emission rates estimates based on line-integrated or point
mole fraction measurements downwind of a defined source.  In this study, we used WindTrax 2.0 (Thunder Beach
Scientific, http://www.thunderbeachscientific.com; Flesch et al., 1995; Flesch et al., 2004). Details on IDM
calculations and resulting $CH_4$ fluxes were presented in You et al. (2020). IDM fluxes of $NH_3$ and total alkane are
shown in this work for comparison with gradient fluxes. Meteorological inputs for calculating inverse dispersion
fluxes of $NH_3$ and total alkane are the same as described in You et al. (2020) for $CH_4$ fluxes.

### 3      Results and discussion

### 3.1  Meteorological conditions

The measurement site including the OP-FTIR was at the south shore of the pond (Fig. S1), therefore the north wind
(wind direction (WD) $\geq$ 286° or WD $\leq$ 76°) was defined as the wind coming from the pond (You et al., 2020). The
wind came from the north for about 22% of the entire measurement period (You et al. (2020) Fig. S1). There was no
significant diurnal variation in wind direction during the study period (You et al. (2020) Fig. S2). Detailed ambient
temperature, water surface temperature, wind speed, and other meteorological parameters can be found in You et al.
(2020). As discussed in You et al. (2020), the warm pond surface resulted in continuing convective turbulence at night,
resulting in continuing transport of pollutants from the pond into the atmosphere without significant diurnal variation.
Gradient and IDM fluxes for $NH_3$ and total alkanes are averages for half hour periods when the wind came from the
pond. The half-hour fluxes were binned into 16 wind direction sectors, and the area-weighted averages of fluxes from
the pond were calculated as described in You et al. (2020).

### 3.2  Methane

Path-integrated mole fractions and associated gradient fluxes of $CH_4$ from OP-FTIR are presented here to test if the
gradient fluxes derived from the mole fractions with OP-FTIR are comparable to $CH_4$ fluxes from eddy covariance
and IDM methods (You et al. (2020)). The area-weighted flux statistics from different methods are summarized in
Table 2. The path-integrated measurement from the FTIR bottom path clearly indicates that the $CH_4$ mole fraction
was elevated when the wind was from the pond direction, while it was steady near 2 ppm when the wind was from


other directions (Fig. S3 and S4). In addition, a clear vertical gradient (Fig. S4), with mole fractions along the bottom path on the order of 0.5 ppm to 1ppm higher than mole fractions from the top path, identified the pond as the $CH_4$ source. The fact that the $CH_4$ mole fraction increased when the wind was from the pond direction, and decreased with height, clearly points to the pond as the dominant local source.

For comparison, vertical profiles of the $CH_4$ mole fraction by point measurements on the nearby tower are given in Supplemental Material Section 2.2. A linear vertical extrapolation of the profiles to the point where the mole fraction reaches 2.0 ppm (background levels) indicated a median plume height of 64m (Fig. S5 and Fig. S6).

The gradient flux derived from the OP-FTIR shows that the flux was minimal when the wind was from other directions, except for the sector centered at 270° (Fig. 1), which represented a mix of pond and shoreline influences).

The average and interquartile ranges of fluxes in wind direction sectors centered at 315°, 337.5° and 0° are comparable. This gradient flux result is consistent with the eddy covariance fluxes measured on the adjacent flux tower (You et al. 2020), and these results also suggest that the pond is the main source of measured $CH_4$ fluxes. However, the sector centered at 292.5° shows average flux 76% and 65% greater than sectors centered at 315° and 337.5°. This is different from the EC fluxes which showed closer agreement between the 292.5°, 315°, 337.5° and 0° sectors (You et al., 2020,

Fig. 7). The footprint of the eddy covariance fluxes measured on the adjacent flux tower at 18m was calculated by using the Flux Footprint Prediction (FFP) model in Kljun et al. (2015), and results showed the 80% contribution distance was typically within 1km which is closer to the main site than the north edge of pond liquid surface (Fig. S1; You et al. (2020)). The outfall was about 1.4 km from the main site. The discrepancy suggests that the footprint of the gradient method incorporated emissions from the outfall more clearly than the smaller footprint of the eddy covariance

method.

FTIR $CH_4$ gradient fluxes and eddy covariance (EC) fluxes showed a linear correlation, but on average, the gradient fluxes were lower than the EC fluxes by 40% ($r^2$=0.56) (Fig. S8). In agreement with EC, the gradient flux showed no significant diurnal variations when the wind was from the pond (Fig. S9, with a relative standard deviation of 36%). To investigate the difference between $CH_4$ gradient fluxes derived from FTIR and EC fluxes, the latter were examined

in relation to meteorological conditions, similar to the analysis presented in You et al. (2020). The analysis in You et al. (2020) showed no correlation between EC flux and friction velocity ($u_*$) or wind speed, while a weak correlation between gradient flux and wind speed is observed (You et al. (2020) Fig. S10). As described in Supplemental Material Section 2.2 (Fig. S5, S6, and S7), $CH_4$ vertical profiles were closer to linear when the wind speed was less than 6 m/s, and were more logarithmic with wind speed greater than 6 m/s. The sector centered at 292.5° was often associated

with wind speeds greater than 6 m/s (21% of the time). The approximation of linear vertical profile could have overestimated the $CH_4$ flux with periods of high winds by 15%. This weak dependence of gradient flux on wind speed and elevated fluxes in sector 292.5° may be partially due to the observed dependence of $CH_4$ vertical profiles on wind speed.

Model calculations by Horst (1999) showed that the estimated footprint of a gradient flux measurement at the

geometric mean height of the gradient is similar to the footprint of EC flux at that same height, for homogeneous upwind area sources. However, mole fraction footprints are significantly larger than perturbation (flux) footprints (Schmid, 1994), and some $CH_4$ sources on the far shore (e.g. the outfall) may have contributed to the upper path $CH_4$



mole fraction. This decreased the vertical gradient difference and thus the derived flux relative to the eddy covariance flux with its smaller footprint, since the latter is more likely to represent water surface emissions only. As an

approximate estimation, the footprint of the path-integrated mole fraction of the top path is about 2.3 km ($23m \times 100$, Flesch et al. (2016)), and this covers the whole pond including the north edge (Fig. S1).

Background mole fractions, upwind of the source under investigation, must be provided for the bLS calculations of $CH_4$ fluxes. We quantified these using two methods. First, the background mole fraction was determined with the FTIR measurements at the south of the pond, as follows: for most of the days, it was taken as the minimum $CH_4$ mole

fraction from the FTIR bottom path on each day while the wind direction was between $180°$ and $240°$. On Aug 7[th] and 30[th], there was no half-hour period when the wind was from this sector, and the background mole fraction was chosen as the minimum mole fraction for the day. For Aug 1[st], there was also no half-hour period for this sector, and the minimum of the day was 2.40 ppm, significantly greater than the minimum mole fraction of other days. Therefore, the background mole fraction of the previous day, 1.92 ppm was used for Aug 1[st].

Alberta Environment and Parks (AEP) conducted OP- FTIR measurements (RAM2000 G2; KASSAY FSI, ITT Corp., Mohrsville, PA, USA) at the north side of the pond (Fig. S1), quantifying $CH_4$ to be used as the second estimate of background mole fractions. For most of the days, the half-hour averaged mole fractions were directly used as the background mole fractions. From Aug 3[rd] 22:00 to Aug 4[th] 13:30, Aug 6[th] 08:00 to 17:00, Aug 23[rd] 1:30 to 2:00, there were no data from AEP, so background mole fractions for these periods were picked as the interpolation of mole

fractions before and after this period. In this approach, the bLS flux results can be negative when the AEP mole fraction is greater than the mole fraction from the measurements at the south shore, possibly due to influences by other emission sources in the surrounding area, gas diffusion under low wind speeds, plume inertia when wind directions changes suddenly, or instrument mismatch differences.

$CH_4$ IDM fluxes with background determined from the first approach (using measurements at the south of the pond)

agreed with IDM fluxes with background determined from the second approach (using measurements at the north of the pond), with a linear regression $r^2$ of 0.92, and a slope of 0.90; there was a 20% difference between average fluxes from the two approaches (Fig. S11; Table S1). These results confirm that the $CH_4$ flux estimate from this inverse dispersion approach is consistent and that the first approach to determining backgrounds is appropriate. In the following results and discussion, IDM fluxes with background mole fractions from the first method are used.

IDM and EC flux showed good comparison (slope=0.93, $r^2$=0.46, You et al., (2020), Fig.8(b)). The interquartile range of the fluxes from these two methods overlap, and the mean IDM fluxes are 11% smaller than EC flux.

### 3.3 $NH_3$

The mole fraction of $NH_3$ was elevated when the wind was from the pond, but was mainly below 5 ppb when the wind was from the south (Fig. 2(a)). $NH_3$ gradient fluxes were significant when the wind came from the pond direction (Fig.

3(a)).

The time series of mole fraction vertical gradient of $NH_3$ and $CH_4$ were similar (Fig. S4). The $NH_3$ gradient flux and $CH_4$ gradient flux showed good correlation ($r^2$=0.8, Fig. 4). The diurnal variation in $NH_3$ gradient flux (relative standard deviation =74%) was stronger than for the $CH_4$ gradient flux (relative standard deviation =36%), with greater


fluxes from 13:00 to 18:00 MDT (Mountain Daylight savings Time) (Fig. S12 (a)). Previous studies showed tailings pond waters contained elevated $NH_3$ concentrations, which makes them potential sources of $NH_3$ to the atmosphere (Allen, 2008; Risacher et al., 2018). $NH_3$ in the pond is mainly produced through nitrate and/or nitrite reduction during microbial activities (Barton and Fauque, 2009; Collins et al., 2016). In addition, some of these nitrate and/or nitrite reduction microbes may also produce reduced sulfur (Barton and Fauque, 2009), and observed reduced sulfur and $NH_3$ from FTIR show good correlation (Moussa et al. 2020a). The water sample collected from Pond2/3 on August 2017 during this study was alkaline (pH = $8.0 \pm 0.5$), which also supports the emission of $NH_3$.

The average (median) flux in the sector centered at 292.5° was 61% (44%) and 73% (28%) more than the average (median) flux in the sectors centered at 315° and 337.5°. These suggest the high average flux in 281-304° is skewed by big spikes which are associated with the outfall (with wind directions in 281-304°), but the majority of $NH_3$ fluxes in the 281-304° wind sector correlated well with $CH_4$ fluxes which were less affected by the outfall. Although hydrotreating processes in upgraders remove most of the sulfur and nitrogen from the bitumen residue, a small amount of $NH_3$ might still be carried with the processed water and tailings (Bytnerowicz et al., 2010), and transported with the outfall liquid into the pond. The negative fluxes observed for the 67.5° sector may be due to elevated $NH_3$ plumes originating from the upgrader facility 3 km upwind in this direction, resulting in a negative gradient and thus deposition to the pond under some circumstances.

IDM fluxes of $NH_3$ were calculated the same way as $CH_4$, and show a weak correlation with $CH_4$ IDM fluxes (Fig. 4b). The $NH_3$ background mole fraction was based on the mean daily minimum, approximately 1ppb (Fig. 2). Vertical profiles of $NH_3$ mole fraction (Fig. S13) with northerly wind also show roughly linear profiles similar to $CH_4$. Profiles of sectors centered at 292.5° and 315° are linear. Therefore, the outfall on average did not significantly contribute to the $NH_3$ profile, i.e. the pond surface was the main source of $NH_3$. $NH_3$ fluxes from the gradient method were significantly less than from the IDM method. This difference is mostly due to the input background $NH_3$ mole fraction. The background $NH_3$ mole fraction was not measured, and could be greater than 1 ppb if there was any source to the north of the pond. Assuming the $NH_3$ gradient flux as a reference, different backgrounds were tested in IDM to match the mean gradient flux. A background of 7 ppb of $NH_3$ was required to close the gap between gradient and IDM fluxes. This seems large but cannot be verified, since there was no ground level measurement of $NH_3$ near the north of the pond. This illustrates the advantage of using either eddy covariance or gradient flux measurements, which are based on mole fraction fluctuations or gradients at a single location and are therefore independent of upwind background mole fraction.

### 3.4 Total alkane

Total alkane derived from FTIR spectra used butane and octane as two surrogates in this study, following the method in EPA OTM10 (Thoma et al., 2010). Results only reflect alkanes which have similar absorption features as butane and octane, and cannot accurately represent other VOCs. Total alkane fluxes from the pond were evident (Fig. 3(b)). A comparison to $CH_4$ fluxes showed only a weak correlation ($r^2$=0.3, Fig. S14), unlike the correlation between $NH_3$ and $CH_4$ (Fig. 4(a)). This difference can be explained by sources of alkane and $CH_4$ at this site. Figure 3(b) shows the average flux from the sector centered at 292.5° is 3.5 g $m^{-2}$ $d^{-1}$, which is 2.43 and 3.33 times of the average fluxes

from the sectors centered at 315° and 337.5°. Figure 1 shows that the average fluxes from the sector centered at 292.5° is 1.76 and 1.64 times of the average fluxes from sectors centered at 315° and 337.5°. These results indicate that there was an enhanced contribution (26%) from around 281° to 304° to total alkane flux measured at the site, but not to the observed $CH_4$ flux. This enhancement of alkane flux is likely due to the outfall, which was at the edge of the pond, 1.9 km from the site at 295°. The liquid mixture flowing into the pond contained naphthenic solvent which include a

mixture of alkanes, alkenes, and aromatic hydrocarbons, and since the outfall is at a temperature of approximately 33° C, enhanced evaporation of volatile components can be expected from this area. The outfall also introduces some mechanical mixing in the upper layers of the pond water, which may contribute to elevated emission rates. The diurnal variation of total alkane gradient flux when the wind came from the pond direction was also weak (the standard deviation of average fluxes at each hour is comparable to the interquartile ranges, Fig. S15). The vertical profiles of

total alkane mole fraction with northeastern winds were vertically invariant (Fig. S16). With northwestern winds, profiles showed a decrease mole fraction from bottom to middle path and minimal decrease or even increase from middle to the top path. These total alkane vertical profiles with northern wind, which are different from $CH_4$ or $NH_3$ profiles, suggesting there were additional sources other than the pond surface to measured total alkane flux, such as the outfall, and industrial activities upwind.

### 3.5 Methanol $CH_3OH$

Unlike pollutants studied above, the $CH_3OH$ mole fraction did not show significant enhancement when wind was from the pond (Fig. S17), suggesting the pond was not significantly contributing to $CH_3OH$ compared to potential sources surrounding the pond. The lifetime of $CH_3OH$ is around 10 days (Simpson et al., 2011; Shephard et al., 2015), and the main source is vegetation (Millet et al., 2008). The mole fractions observed at the site were consistent with satellite

measurements representative of the general oil sand region (Shephard et al., 2015), and with an airborne study of VOCs (Simpson et al., 2010). In addition to emissions from vegetation, $CH_3OH$ observed in the oil sands region could be due to transport from biomass burning (Simpson et al., 2011; Bari and Kindzierski, 2018), and local traffic (Rogers et al., 2006; You et al., 2017).

### 3.6 Comparison of calculated fluxes to reported emissions and approaches

As a test of the robustness of these results, fluxes of $CH_4$, $NH_3$ and total alkane are also calculated from the "slant path" method described by Flesch et al. (2016). Calculation inputs and results are summarized in the Supplemental Material Section 6. Compared to gradient flux results with our modified Bowen ratio approach, $CH_4$, $NH_3$, and total alkane fluxes with the "slant path" flux-gradient method were 27%, 40%, and 56% smaller (Table S1). The difference in fluxes from the two approaches could be due to differences in assumptions regarding the vertical profiles. In our

modified Bowen ratio gradient flux approach, we used linear vertical mole fraction profiles of pollutants to calculate the difference of mean heights between the two paths. In Flesch et al. (2016) the gradient flux for pairs of points along the two paths was integrated along the entire path length assuming the flux was uniform horizontally. In that approach, the dependence of $K_c$ on height is incorporated explicitly, assuming a logarithmic wind profile including a stability





correction. In our modified Bowen ratio approach, $K_c$ is derived from a measured and stability corrected $K_m$, and therefore did not require a wind profile shape assumption.

To facilitate a transparent comparison of the emission results from this study to reported facility wide emissions of Suncor, we present emission rates based on simple extrapolation of the measured August emissions to the whole year. Other possible seasonal emission profiles have been reported (Small et al., 2015); using these to convert from August emissions to annual average values would for example result in a scale factor of 0.92 (mass transfer model), 0.64 (mass transfer model adjusted for ice cover), or 0.42 (thawing degree-day model) (Cumulative Environmental Management Association, 2011). The seasonally invariant total emission estimate for $NH_3$ and total alkanes from Pond 2/3 to the air were 92 and 1120 tonnes $y^{-1}$ in 2017. However, $NH_3$ emissions from Pond 2/3 have not been reported in the past, because $NH_3$ was not being measured as part of compliance flux chamber monitoring. Therefore, the facility wide $NH_3$ emissions reported to the Government of Canada National Pollutant Release Inventory (NPRI) (0.82 tonnes $y^{-1}$) in 2017 did not include fugitive emissions from tailings ponds. The solvents entering this tailings pond are naphtha additives with octane, nonane and heptane as the biggest contributors. Li et al. (2017) quantified alkane emissions (including n-alkanes, branched-alkanes, and cycloalkanes) as 36.2 tonne $d^{-1}$ from the whole facility using airborne measurements in 2013, which is 73.9% of 49 tonne $d^{-1}$ total VOC emission. Based on the VOC profile shown in Moussa et al. (2020b), heptane and octane were the highest emitted VOCs, and alkanes account for 54% of total VOC emitted from Pond2/3 in 2017. If we use reported facility wide VOC annual emissions in 2017 (17242 tonnes $y^{-1}$, NPRI) to estimate facility wide total alkane annual emission, we obtain $17242 \times 53\% = 9138$ tonnes $y^{-1}$, and total alkane emissions from Pond2/3 contribute 12.2% to facility wide emissions for 2017. The fugitive $NH_3$ emissions from Pond2/3 in this study were 92 tonnes $y^{-1}$, a number that is 113 times the process related emission number reported for the facility to NPRI. Negligible volume of ammonia may be carried over to the pond through the naphtha recovery unit and FTT line and it is believed that the ammonia is mainly generated from the biogenic activities in the MFT layer of the pond. The majority of $H_2S$, $NH_3$ and $CH_4$ emissions are related to microbiological activities as evidenced in this study.

## 4    Conclusions and Implications

We have shown that OP-FTIR is an effective method to quantify mole fractions and vertical gradients of $CH_4$, $NH_3$, and total alkanes continuously and simultaneously for an area source such as a tailings pond. Benefits are the integration of mole fractions over long path lengths, thus providing a spatially representative average, and the avoidance of sample line issues that can be serious problems for sticky gases such as $NH_3$. Results from gradient (modified Bowen ratio) method and IDM calculations suggest OP-FTIR is a useful tool for deriving emission rates of $CH_4$, $NH_3$ and total alkane from this type of fugitive area source. For the two approaches of determining background mole fractions of $CH_4$ with the IDM method, i.e., upwind background measurement vs. local background estimation, the area weight-averaged fluxes of $CH_4$ were within 20%. FTIR $CH_4$ gradient fluxes and EC fluxes showed a linear correlation, but on average, the gradient fluxes were lower than the EC fluxes by 40%. IDM and EC flux showed good comparison, and the mean IDM fluxes are 11% smaller than EC flux. $NH_3$ gradient flux and IDM flux showed a difference of more than 50%, which suggests that there may have been sources of $NH_3$ upwind (north) of the pond


that were not captured by assuming that southern and northern background $NH_3$ were similar, thus illustrating a limitation of the IDM method. $CH_4$, $NH_3$, and total alkane fluxes were also calculated using the "slant path" flux-gradient method (Flesch et al. 2016), to compare to the modified Bowen ratio approach, and results were 27%, 40%, and 56% lower than the modified Bowen ratio approach.

The $NH_3$ emissions results in this study are the first to quantify $NH_3$ fugitive fluxes from a tailings pond and clearly

showed that Pond2/3 is a significant source of $NH_3$, most likely through microbial activities in the pond. This suggests that at least some tailings ponds in the oil sands could be significant sources of $NH_3$, compared to process-related facility emissions. Further measurements of $NH_3$ emissions from tailings ponds are recommended to elucidate our understanding of the mechanisms behind $NH_3$ emissions and to improve the total facility emission estimates reported to NPRI.

Total alkane gradient fluxes from OP-FTIR measurements clearly showed that the pond is a significant source of total alkane. Annual alkane emissions extrapolated from these measurements represented 12.2% of facility emissions. The outfall area contributed significantly (26%) to pond alkane emissions, showing spatial variability of alkane emissions from the pond. Observed $CH_3OH$ mole fractions show that the pond was unlikely a significant source of $CH_3OH$. This study demonstrated the applicability of OP-FTIR combined with modified Bowen ratio or inverse dispersion methods

for determining emission fluxes of multiple gases simultaneously, with high temporal resolution and comprehensive spatial coverage.

**Data availability.**

All data are publicly available at http://data.ec.gc.ca/data/air/monitor/source-emissions-monitoring-oil-sands-

region/emissions-from-tailings-ponds-to-the-atmosphere-oil-sands-region/.

**Author contributions.**

YY and RS wrote the manuscript; SM, LZ, LF and JB contributed data and comments.

**Competing interests.**

Dr. Beck is an employee of Suncor Energy. The other authors have no competing interests.

**Acknowledgements.**

The authors thank the technical team of Andrew Sheppard, Roman Tiuliugenev, Raymon Atienza and Raj Santhaneswaran for their invaluable contributions throughout, Julie Narayan for spatial analysis, Stewart Cober for management and Stoyka Netcheva for home base logistical support. We thank Suncor and its project team (Dan Burt et al.), AECOM (April Kliachik, Peter Tkalec) and SGS (Nathan Grey, Ardan Ross) for site logistics support. This

work was partially funded under the Oil Sands Monitoring Program and is a contribution to the program but does not necessarily reflect the position of the program. We also acknowledge funding from the Program for Energy Research and Development (Natural Resources Canada) and from the Climate Change and Air Pollution Program (ECCC). The works published in this journal are distributed under the Creative Commons Attribution 4.0 License. This licence does not affect the Crown copyright work, which is re-usable under the Open Government Licence (OGL). The Creative


Commons Attribution 4.0 License and the OGL are interoperable and do not conflict with, reduce or limit each other.
© Crown copyright 2020

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



**Tables**

Table 1 Spectral windows of OP- FTIR spectra for retrieving mole fractions of pollutants in this study.

| Pollutant name | Chemical formula | Spectral Window (cm$^{-1}$) | Interference gases | Threshold correlation coefficient[a] | Detection limit[c] | Paths |
|---|---|---|---|---|---|---|
| Methane | $CH_4$ | 3006-3021 | $H_2O$ | 0.95 | 1.1 ppb | All three |
| Ammonia | $NH_3$ | 957-973 | $H_2O$, $CO_2$ | 0.3 | 1.1 ppb | All three |
| Methanol | $CH_3OH$ | 1020-1040 | $H_2O$, $NH_3$, $O_3$, $C_2H_5OH$, $C_6H_6$ | 0.3 | 1.1 ppb | All three |
| Butane[b] | n-$C_4H_{10}$ | 2804-3001 | $H_2O$, $CH_4$, $CH_3OH$, HCHO, n-$C_7H_{16}$, n-$C_6H_{14}$, n-$C_8H_{18}$, $CH_3CH(CH_3)C_3H_7$ | 0.1 | 1.1 ppb | All three |
| Octane[b] | n-$C_8H_{18}$ | 2804-3001 | $H_2O$, $CH_4$, $CH_3OH$, HCHO, n-$C_7H_{16}$, n-$C_6H_{14}$, $CH_3CH(CH_3)C_3H_7$, $C_2H_5CH(CH_3)C_2H_5$ | 0.1 | 0.9 ppb | All three |
| Formaldehyde | HCHO | 2730-2800 | $H_2O$, $CO_2$, $CH_4$ | 0.2 | 2.3 ppb | Bottom only |
| Carbon dioxide | $CO_2$ | 2030-2133 | $H_2O$, CO | 0.8 | - | Bottom only |


[a] Threshold correlation coefficient is a input for OPUS_RS when performing fitting analysis of FTIR spectra. When the correlation coefficient between measured spectrum and reference spectrum with the defined spectral window is below this threshold, that pollutant is not "identified" and the mole fraction is reported as zero in OPUS_RS (You et al., 2017).

[b] Butane and octane mixing ratio are quantified as two surrogates to quantify a "total alkane" mixing ratio = Butane + Octane (Thoma et al., 2010).

[c] Detection limit is calculated by converting 3σ of the noise of the measurements with a retroreflector distance of 225m by Bruker to 3σ of the noise with 200m in this study.





Table 2 Summary of fluxes from OP-FTIR measurements. Results are area weight-averaged fluxes from the pond.

| All fluxes in g m$^{-2}$d$^{-1}$ | Flux method | Q_25% | Median | Q_75% | Mean[a] |
|---|---|---|---|---|---|
| CH$_4$ | Tower EC | 4.2 | 5.9 | 7.9 | 6.1± 0.5 |
| | FTIR gradient | 1.9 | 3.4 | 5.5 | 3.7 ± 0.5 |
| | IDM | 3.6 | 5.2 | 6.6 | 5.4± 0.4 |
| NH$_3$ | gradient | 0.01 | 0.04 | 0.08 | 0.05 ± 0.01 |
| | IDM | 0.06 | 0.09 | 0.15 | 0.11 ± 0.01 |
| Total alkane | gradient | 0.25 | 0.95 | 1.97 | 1.33 ± 0.19 |
| | IDM | 0.57 | 0.94 | 1.56 | 1.33 ± 0.10 |

[a] Errors with the mean fluxes are calculated with a "top-down" approach: the average of standard deviations of fluxes from five periods when the fluxes were relatively steady.





**Figures**

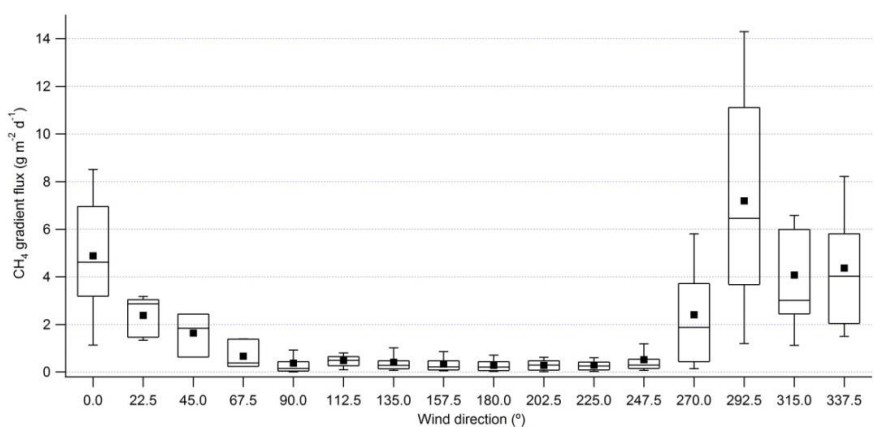

**Figure 1 Gradient flux of CH4 from FTIR binned by wind direction in 22.5-degree bins**



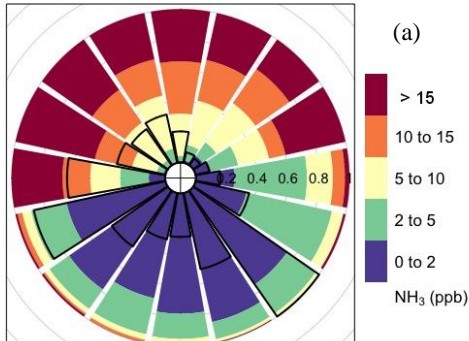


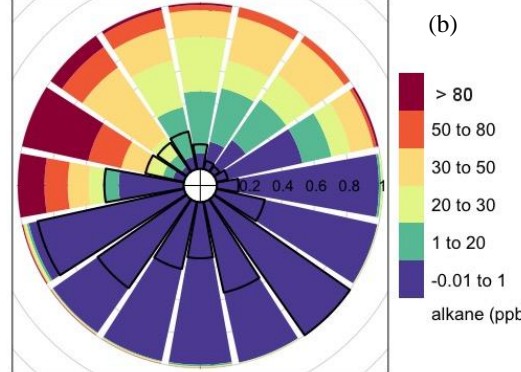

**Figure 2 Normalised rose plot of NH₃ (a) and total alkane (b) mole fractions from FTIR bottom path. Colors represent mole fraction in ppb. The length of each colored segment presents the time fractions of that mole fraction range in each direction bin. The radius of the black open sectors indicates the frequency of wind in each direction bin; angle represents wind direction.**






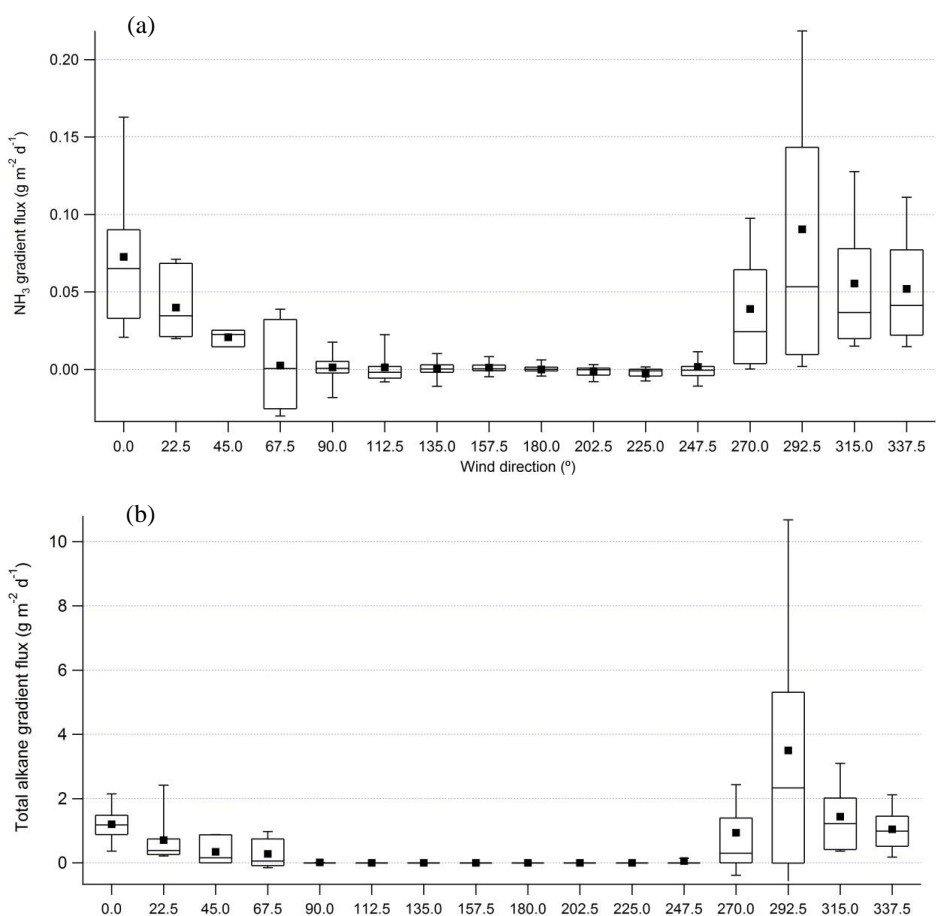

**Figure 3 Gradient flux of NH₃ (a) and total alkane (b) from FTIR top-bottom path binned by wind direction in 22.5-degree bins.**





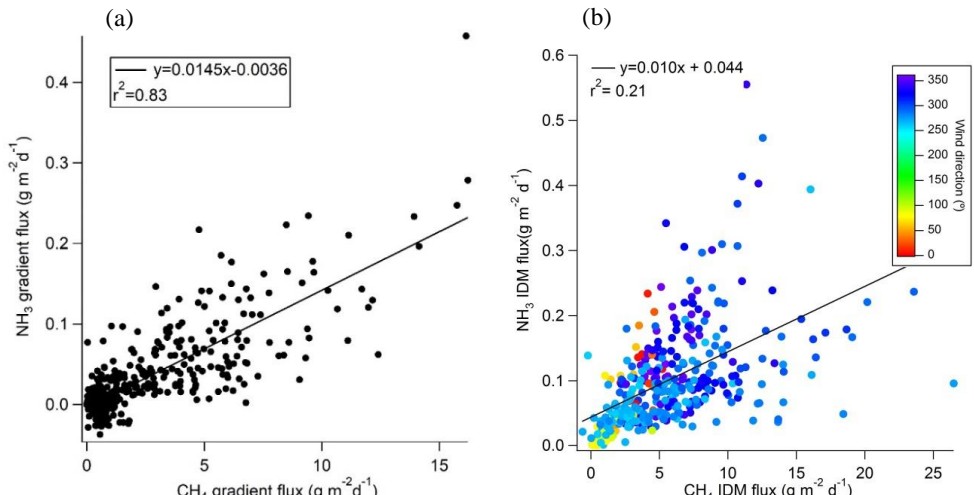

**Figure 4 (a): NH$_3$ gradient flux compared to CH$_4$ gradient flux; (b): NH$_3$ IDM flux compared to CH$_4$ IDM flux**