# Peer review of "Quantifying fugitive gas emissions from an oil sands tailings pond with open-path FTIR measurements"

_Atmospheric Measurement Techniques, 2020_

## Referee Comment (RC1) · Anonymous Referee #1 · 8 Sep 2020

Review of "Quantifying fugitive gas emissions from an oil sands tailings pond with open-path FTIR measurements" by You et al., submitted for publication in Atmospheric Measurement Techniques (amt-2020-257). This manuscript presents results from a field study conducted in the oil sands region of Canada, where gas emissions of CH4, NH3, and alkanes were measured from a tailings pond using micrometeorological techniques.

General Comments

The strength of this work is the scientific significance of the emission measurements that have been documented. The subject matter is of broad interest due to the large scale of industrial operations in the oil sands region of Canada, the use of large tailings ponds in these operations, and a paucity of information on gas emissions from these

sites. However, the scientific quality of the work is mixed. Several methodological details need to be clarified. And while some aspects of the study are well-explored, others are not given sufficient attention. Finally, the overall presentation of the manuscript should be improved. Basic information that should be in the manuscript is missing and the reader is forced to look to a companion paper and to supplemental material for that information. It is difficult for the reader to understand this paper without a good deal of effort. My overall recommendation is that the flux calculation details need to be significantly clarified and documented within the manuscript. This work could then make a valuable contribution to our understanding of the environmental footprint of oil sands operation.

Specific Comments

1) Questions related to FTIR measurements. I have two questions regarding the FTIR concentration retrievals and the related flux calculation.

a. The authors state (line 108-109) that "... temperature and pressure dependent reference files were used for fitting and retrieving mole fractions." It is unclear how this was done. My concern is that with vertical temperature gradients at the measurement site (as implied by Fig. 4 in the companion paper by You et al.), different temperatures should be used for retrievals for the different FTIR path heights. If not done, is there potential for "false" flux signals? The authors should comment on this.

b. The second question regards the concentration measurement used to calculate flux in Eq. 1. Following the classic WPL flux corrections for flux-gradient formulae (Webb, et al. 1980. Quart. J. R. Meteorol. Soc. 106, 85–100), should the concentrations be the mole fraction with respect to dry air (mixing ratio)? This should be clarified.

2) The exact procedure for calculating the FTIR fluxes is unclear. In one section the authors indicate the critical tracer diffusivity (Kc) for the FTIR calculations was extrapolated from Kc measured from point concentrations (Eq. 2), and at another point they state that Kc is determined from the measured eddy diffusivity Km (Line 314). Looking

over this paper, the companion paper, and the supplemental material, I am uncertain as to what was done. The methodology needs to be better explained.

3) I am concerned about how tracer diffusivity (Kc) is calculated in Eq (2). The calculation assumes a linear change in Kc with height, but the general view is that diffusivities (heat, momentum, moisture) are non-linear except in neutral stratification. This may have an impact on the calculated fluxes. Consider the standard definition of Km:

. $Km = k\_v * ustar * z / PHI\_m$,

where PHI_m is the non-dimensional wind shear. A commonly used PHI_m relationship is given by Hogstrom (1996) for unstable conditions:

. $PHI\_m = (1 − 19*z/L) ˆ -0.25$

If we redo the calculation outlined in the manuscript Eq. (2) to determine Km for the FTIR gradient (z = 1, 12 m) using the Km calculated from the point measurements (z = 8, 32 m), with the above expressions and assuming L = - 20 m, then Km_FTIR/Km = 0.25. This is 25% lower than the 0.325 value the authors calculated assuming a linear relationship. This implies the fluxes calculated by the authors may overestimate the fluxes in unstable conditions. While this example is for the case of Km, one would expect a similar non-linear relationship for Kc.

4) There should be a deeper discussion of the relationship between the different flux measurement techniques. I have four questions/concerns in this regard.

a. How much are the flux-gradient (FG) measurements constrained to match the eddy covariance (EC) measurements? It strikes me that calculating the tracer diffusivity (Kc) from the concentration gradient and the EC flux, and then using that Kc in the FG calculation will act to force the FG and EC fluxes to be equivalent. Yet in several places (lines 180-210) the authors discuss how the FG and EC fluxes are different. How do we reconcile those two things?

b. As mentioned before, it is unclear how the FTIR fluxes were calculated in this study.

Based on the companion paper, I suspect the turbulent Schmidt number (Sc) was used to estimate Kc from the eddy diffusivity Km (in the companion paper the authors give an interesting evaluation of Sc and conclude that over a broad stability range Sc = 0.923). A conclusion of this manuscript is that the FG fluxes from the pond are 40% lower than the EC fluxes (Line 192). This difference between FG and EC could be erased with a smaller Sc (for which there is good evidence in the literature). In looking at the CH4 fluxes from the three different measurement techniques (Table 2), could the average from all three be statistically identical if Sc = 0.6 is used in the FG calculations?

c) The authors discuss the difference in fluxes from the FG and the EC techniques in terms of their different measurement footprints. I am not convinced the difference is large. The authors state that the mole fraction footprint (FG) is larger than the flux (EC) footprint (Line 206). It is true that a concentration footprint is much larger than the corresponding flux footprint, but an FG footprint is given by a difference in two concentration footprints (two heights), which is not so different from the flux footprint (if the two FG heights are not too far apart). In other words, a distant source contributes almost exactly the same concentration at both the top and the bottom measurement heights, so the footprint difference for distant sources is zero. I think the difference between the FG and EC measurements may have another explanation.

d) The authors compare the fluxes calculated using their FG formula with those using the slant-path FG formula from Flesch et al. (2016), and found the slant-path calculations give fluxes that are 27 to 56% lower. This result is difficult to understand, given that both calculations start with the same underlying FG calculation approach and use the same concentration gradients. The difference could be due to the assumed Sc, or how the path integration of the FTIR concentration and Kc is handled. There is a difference in Sc between the two calculations. But because the Sc used in the slant-path formula (0.64) is smaller than used in this study ($\sim$0.923), the slant path flux magnitudes should be higher, not lower. I suspect the difference is related to the diffusivity calculation discussed in comment 3 above. The authors should explain how these two

different FG formula can lead to such different results. This is important for confidence in their FG methodology, as the slant-path formula is a more rigorous expression of the FG relationship for path-integrated measurements than their working formula.

Technical Corrections

Line 75: it is unclear what is meant by "turbulent fluxes" in this context. Gas fluxes?

Line 85: "Other experimental details of the project can be found in You et al. (2020)". This is one of several places in the manuscript where critical pieces of information are missing and should be included: e.g., a figure showing the equipment locations with respect to the tailings pond. I noted that Lines 128, 150, and 160 refer the reader to critical details in You et al. (2020). I would like to see some of that other-source material moved into this manuscript.

Line 117. "For these trace gases at this site, the detection limits of this open path system were insufficient . . .". This is very interesting.

Line 122. Does the flux calculation really fit the usual description of a modified bowen ratio (MBR) method? It's difficult to know because of uncertainty as to how the fluxes are calculated. Usually an MBR measurement means the unknown flux of the gas of interest is related to the known flux of another tracer gas, plus the gradient of that tracer gas, and the gradient of the gas of interest . No theoretical flux gradient relationships are needed. In this study, it is likely that the tracer diffusivity $K_c$ is estimated from a stability corrected $K_m$ measured with a sonic anemometer, I would say the technique is better described as an aerodynamic flux-gradient approach.

Line 160: ". . . the warm pond surface . . . resulting in continuing transport of pollutants . . . without significant diurnal variation." When the pond is warm relative to the air, does this mean there are no diurnal changes in atmospheric stability over the pond? I doubt it. This statement also implies that aerodynamic resistance is the limiting factor controlling pond emissions.

Line 314: "In our modified Bowen ratio approach, Kc is derived from a measured and stability corrected Km . . .". This sentence highlights the confusing description of the methods used in this paper: I do not think Km has been defined to this point in this manuscript; and Eq. (2) suggests that Kc for the FTIR is calculated from Kc taken from point measurements (and not from Km).

―――――――――――――――――――

---

## Referee Comment (RC2) · Anonymous Referee #2 · 19 Sep 2020

This manuscript is difficult to review. Some key materials are in the supplementary file but that file resembles more an internal research note than a polished document for publication. Further complicated my reading is the fact that this manuscript overlaps, in terms of topic, methods and data, with another AMT manuscript currently under review. I felt like I were actually reviewing three papers instead of one. It took me a lot of effort to piece together a storyline from the information scattered across these three documents. For this reason, I recommend that the authors undertake a complete rewrite, with the aim of producing a stand-alone, coherent paper.

The open-path FTIR is subject to density effects due to vertical temperature and humidity gradients. Because they are stronger for gases of lower concentration, ratio-ing the uncorrected molar concentration gradients will not eliminate these effects. I am not

sure that I trust their flux values without correcting for these effects.

You seem to rely on eddy-covariance methane flux and concurrent measurement of methane concentration gradient to obtain fluxes of other trace gases from the modified Bowen ratio method. How did you get the gradient CH4 flux then? The gradient CH4 flux was biased low in comparison with the eddy-covariance flux. Were other fluxes similarly biased (due to a limited fetch). Can you estimate the "true" emission fluxes of the tailings pond via footprint modeling?

The section on methanol CH3OH should be enhanced. What was the average flux? Did the flux vary with environmental conditions?

The section on comparison with published fluxes is a bit superficial. The reader is interested in knowing if your emission numbers are representative of a typical tailings pond. Also a solid comparison will require footprint correction to your gradient fluxes.

---

## Author Comment (AC1) · 30 Oct 2020

**Response to Referee #1, "Quantifying fugitive gas emissions from an oil sands tailings pond with open-path FTIR measurements"**

**We thank the referee for this thorough review. We have carefully considered all the comments, implemented many of the suggestions, and included more details on methodology to make this manuscript stronger and stand-alone. Below, we address each question in turn. Questions and suggestions are in black, and our responses are in blue.**

Specific comments:

1) Questions related to FTIR measurements. I have two questions regarding the FTIR

concentration retrievals and the related flux calculation.

a.        The authors state (line 108-109) that ". . . temperature and pressure dependent reference files were used for fitting and retrieving mole fractions." It is unclear how this was done. My concern is that with vertical temperature gradients at the measurement site (as implied by Fig. 4 in the companion paper by You et al.), different temperatures should be used for retrievals for the different FTIR path heights. If not done, is there potential for "false" flux signals? The authors should comment on this.

Response: The temperature measured at 8m was used in retrieving mole fractions for the three paths. The temperature difference shown in You et al. (2020), Figure 4c, is the difference between the pond liquid surface and the temperature measured at 8m, which was usually bigger than the difference between the temperature at 8m and the actual mean temperature for each path. To address this question, we have compared the temperature at 8m to temperature at 1m, and temperature at 18m to temperature at 8m. The statistics are shown here:

[Figure]

**Figure: Histogram of (a) T_8m – T_1m (K) and (b) T_18m – T_8m (K) over the entire study period.**

When the wind was from the pond, (T_8m – T_1m) was mainly between – 1.5 and 0.5 K, and (T_18m – T_8m) was mainly between -0.25 and 0.25 K.

To improve on the method we employed in our original manuscript, we can use T_1m for the bottom path, and an interpolation of T_8m and T_18m at 12m for the top path, to calculate mole fractions. First, the difference between the T_8m (which was used in previous retrieval for all the paths) and T_1m, as well as the difference between T_8m and T_12m, was calculated. In You et al. (2017), the sensitivity of the input temperature on the retrieved mole fractions of $CH_4$ and $NH_3$ was investigated using the same

software (OPUS_RS). Over a 45-degree range (from 278 to 323K), the retrieved mole fraction time series changed by less than 4.2% for $CH_4$ and by less than 8.9% for $NH_3$. We assume that the sensitivity of temperature on retrieved mole fractions of alkanes is similar to $CH_4$, due to the similar absorption mechanisms. The retrieved mole fractions using T_8m (x_original) were corrected as follows:

Bottom path:  x_corrected = x_original  + x_original × (T_1m – T_8m)/45 × 4.2%

Top path:  x_corrected = x_original  + x_original × (T_12m – T_8m)/45 × 4.2%

For $NH_3$, the 4.2% is replaced by 8.9%.

Next, the $H_2O$ mole fraction was used to calculate dry mole fractions. Finally, the same gradient flux calculations were performed (using the original Kc-FTIR) with these new temperature corrected dry mole fractions for $CH_4$, $NH_3$ and total alkanes. Results are shown in the Table 1 below. The relative changes in the final fluxes are 8%, 0%, and -25%, for $CH_4$, $NH_3$ and total alkanes respectively. While these changes do not affect the main results and conclusions in this paper, they are significant enough that we included them in our flux calculations in the revised manuscript. Note that the numbers in Table 1 below are different from those presented in the revised manuscript, since the latter incorporate other modifications in response to question 3) below.

Table 1: Statistics of gradient fluxes from FTIR with different processes.

| | Flux (g m$^{-2}$ d$^{-1}$) | q_25% | median | q_75% | mean |
|---|---|---|---|---|---|
| CH$_4$ | Original | 1.9 | 3.4 | 5.5 | 3.7 |
| | Dry correction | 2.1 | 3.3 | 5.6 | 3.9 |
| | T correction + Dry correction | 2.1 | 3.3 | 5.6 | 4.0 |
| NH$_3$ | Original | 0.01 | 0.04 | 0.08 | 0.05 |
| | Dry correction | 0.02 | 0.04 | 0.08 | 0.05 |
| | T correction + Dry correction | 0.02 | 0.04 | 0.08 | 0.05 |
| Total alkanes | Original | 0.25 | 0.95 | 1.97 | 1.33 |
| | Dry correction | 0.23 | 0.66 | 1.55 | 1.00 |
| | T correction + Dry correction | 0.23 | 0.67 | 1.55 | 1.00 |

**b.      The second question regards the concentration measurement used to calculate flux in Eq. 1. Following the classic WPL flux corrections for flux-gradient formulae (Webb, et al. 1980. Quart. J. R. Meteorol. Soc. 106, 85–100), should the concentrations be the mole fraction with respect to dry air (mixing ratio)? This should be clarified.**

Response: We appreciate the reviewer pointing out this. Yes, the concentration used for flux calculations should properly be the mole fraction with respect to dry air. To investigate how much of a difference this makes to the final gradient fluxes, we recalculated the fluxes using dry mole fractions. The temperature and relative humidity at 1m, 8m and 18m were used to calculate $H_2O$ mole fractions at these heights, respectively. The calculated $H_2O$ mole fraction at 1m was used to calculate dry mole faction of gases for the bottom path. The average of the calculated $H_2O$ mole fractions at 8m and 18m was used to calculate dry mole fraction of gases for the top path (with an average height of 12m). The calculated dry mole

fraction of $CH_4$ for the bottom path was calibrated with the CRDS $CH_4$ dry mole fraction at 4m, with the exactly same process as illustrated in Figure S1. After calculating the temperature corrections discussed in point 1 (a) above, the last steps were calculating dry mole fractions and new gradient fluxes, as described in point 1 (a) above.

2) The exact procedure for calculating the FTIR fluxes is unclear. In one section the authors indicate the critical tracer diffusivity (Kc) for the FTIR calculations was extrapolated from Kc measured from point concentrations (Eq. 2), and at another point they state that Kc is determined from the measured eddy diffusivity Km (Line 314). Looking over this paper, the companion paper, and the supplemental material, I am uncertain as to what was done. The methodology needs to be better explained.

Response: We thank both reviewers for pointing out that more text was needed in this manuscript to make it stand on its own. We have added substantial material to Section 2 on the measurement details and calculation methodology.

3) I am concerned about how tracer diffusivity (Kc) is calculated in Eq (2). The calculation assumes a linear change in Kc with height, but the general view is that diffusivities (heat, momentum, moisture) are non-linear except in neutral stratification. This may have an impact on the calculated fluxes. Consider the standard definition of Km:
Km = k_v * ustar * z / PHI_m,
where PHI_m is the non-dimensional wind shear. A commonly used PHI_m relationship is given by Hogstrom (1996) for unstable conditions:
PHI_m = (1 – 19*z/L) ^ -0.25
If we redo the calculation outlined in the manuscript Eq. (2) to determine Km for the FTIR gradient (z = 1, 12 m) using the Km calculated from the point measurements (z = 8, 32 m), with the above expressions and assuming L = - 20 m, then Km_FTIR/Km = 0.25. This is 25% lower than the 0.325 value the authors calculated assuming a linear relationship. This implies the fluxes calculated by the authors may overestimate the fluxes in unstable conditions. While this example is for the case of Km, one would expect a similar non-linear relationship for Kc.

Response: We thank the reviewer for pointing this out. We have recalculated the fluxes using a stability-dependent ratio of Km_FTIR/Km. Details are provided in the revised manuscript. The new mole fraction gradients are the results after temperature effect and dry mole fraction calculations, as discussed in responses for the first two comments above. The new gradient fluxes, incorporating the temperature and density effects discussed under question 1) as well as the new eddy diffusivity scaling factors, are less than the original fluxes, by 8%, 20%, and 37% for $CH_4$, $NH_3$ and total alkanes respectively.

4) a. How much are the flux-gradient (FG) measurements constrained to match the eddy covariance (EC) measurements? It strikes me that calculating the tracer diffusivity (Kc) from the concentration gradient and the EC flux, and then using that Kc in the FG calculation will act to force the FG and EC fluxes to be equivalent. Yet in several places (lines 180-210) the authors discuss how the FG and EC fluxes are different. How do we reconcile those two things?

Response: Yes, Kc for 8m and 32m was indeed derived from combining the $CH_4$ EC flux and $CH_4$ gas vertical gradients as measured on the EC tower, as described in detail in You et al. (2020). The difference between that paper and this manuscript is that here the gradient fluxes were from the FTIR top and bottom path as opposed to from the tower measurements. Differences and uncertainties arise from the vertical profile of Kc as discussed under question 3 above), as well as from the vertical profile of the $CH_4$ mole fraction, investigated in detail in the Supplement.

b. As mentioned before, it is unclear how the FTIR fluxes were calculated in this study. Based on the companion paper, I suspect the turbulent Schmidt number (Sc) was used to estimate Kc from the eddy diffusivity Km (in the companion paper the authors give an interesting evaluation of Sc and conclude that over a broad stability range Sc = 0.923). A conclusion of this manuscript is that the FG fluxes from the pond are 40% lower than the EC fluxes (Line 192). This difference between FG and EC could be erased with a smaller Sc (for which there is good evidence in the literature). In looking at the CH4 fluxes from the three different measurement techniques (Table 2), could the average from all three be statistically identical if Sc = 0.6 is used in the FG calculations?

Response: Using Sc = 0.6, the mean gradient flux from the tower (8m, 32m) would be 7.4 $gm^{-2}d^{-1}$, statistically not different from the mean EC flux. Assuming Sc = 0.6 and the original Kc vertical scales, the mean gradient flux from FTIR top-bottom paths would be 4.8 $gm^{-2}d^{-1}$, which is also statistically equivalent to the mean EC flux.

However, based on our literature search on atmospheric Schmidt numbers, we felt that there is no strong consensus on their value. Flesch (2002) presented Sc results that covered a broad range from 0.17 to 1.34. Gualtieri et al.(2017) showed Sc values from previous experimental and numerical simulations from 0.1 to 1.3. Based on our own tower flux gradient measurements, described in You et al (2020), we found that Sc varied from 0.04 to 3.26 as a function of stability. In order to make the tower EC and gradient fluxes mutually consistent, the calculated Sc=0.923 (neutral) needs to be used. Since these two measurements were collocated on the same tower, it is reasonable to assume that they should result in the same flux. The comparison to the FTIR gradient fluxes is not as direct due to differing measurement geometries and slight lateral displacement, which may explain some of the differences in the calculated fluxes.

c. The authors discuss the difference in fluxes from the FG and the EC techniques in terms of their different measurement footprints. I am not convinced the difference is large. The authors state that the mole fraction footprint (FG) is larger than the flux (EC) footprint (Line 206). It is true that a concentration footprint is much larger than the corresponding flux footprint, but an FG footprint is given by a difference in two concentration footprints (two heights), which is not so different from the flux footprint (if the two FG heights are not too far apart). In other words, a distant source contributes almost exactly the same concentration at both the top and the bottom measurement heights, so the footprint difference for distant sources is zero. I think the difference between the FG and EC measurements may have another explanation.

Response: We completely agree with the reviewer that for locations with very large homogeneous fetches, EC footprints and gradient footprints with equivalent geometric mean heights are indeed equivalent. In our case, however, the median footprint in northern sectors was under most stability conditions just slightly smaller than the pond, which means that the upper gradient level was more likely to see non-pond influences. Another potential reason for differences is that as seen in Fig. 1, the FTIR path was offset by an average of about 100m to the east relative to the EC tower, and integrated over a path with some horizontal extent (200m). If the pond had less homogeneous in terms of fugitive emissions than generally assumed, there may be a difference in fluxes even over such a small distance.

d) The authors compare the fluxes calculated using their FG formula with those using the slant-path FG formula from Flesch et al. (2016), and found the slant-path calculations give fluxes that are 27 to 56% lower. This result is difficult to understand, given that both calculations start with the same underlying FG

calculation approach and use the same concentration gradients. The difference could be due to the assumed Sc, or how the path integration of the FTIR concentration and Kc is handled. There is a difference in Sc between the two calculations. But because the Sc used in the slant-path formula (0.64) is smaller than used in this study (0.923), the slant path flux magnitudes should be higher, not lower. I suspect the difference is related to the diffusivity calculation discussed in comment 3 above. The authors should explain how these two different FG formula can lead to such different results. This is important for confidence in their FG methodology, as the slant-path formula is a more rigorous expression of the FG relationship for path-integrated measurements than their working formula.

Response: We thank the reviewer for this thoughtful question. Yes, the slant-path and our calculations both start with the same gradient flux equation containing Kc = Km/Sc. The two calculations become different when calculating Km. In this study, Km was calculated from the measured momentum flux and measured wind speed vertical gradient $\Delta u/\Delta z$ between 8m and 32m. This detail has now been inserted into the manuscript. In Flesch et al. (2016), $\Delta u$ was calculated in terms of the stability corrected log-wind profile in their Eq (5) and (6), as explained in Section 3.6. There was no other difference in the two approaches. We stated in the Supplement, Section 5, that "In this study, calculated $S_c$ is allowed to vary with dynamic stability (You et al. (2020) Fig. 3), while in Flesch et al. (2016) $S_c$ was a constant 0.64." To evaluate this, in the current version of the manuscript we used equation (9) from Flesch et al. (2016) in comparison with our gradient approach using a variable Sc in equation (9). So in the two approaches presented in this study, the exactly same time series of Sc were used.

In addition, this Table S2 was also revised. The flux calculation was the same as the original fluxes. The difference is the input mole fraction gradient changed after correction temperature and dry mole fractions. The recalculated results show reasonable agreement between the slant-path and our gradient fluxes (differences are within 30% in Section 3.6).

Technical Corrections

Line 75: it is unclear what is meant by "turbulent fluxes" in this context. Gas fluxes?

Response: We meant sensible heat fluxes and momentum fluxes. "Turbulent fluxes" are now changed to "sensible heat fluxes and momentum fluxes".

Line 85: "Other experimental details of the project can be found in You et al. (2020)". This is one of several places in the manuscript where critical pieces of information are missing and should be included: e.g., a figure showing the equipment locations with respect to the tailings pond. I noted that Lines 128, 150, and 160 refer the reader to critical details in You et al. (2020). I would like to see some of that other-source material moved into this manuscript.

Response: We appreciate this comment. The revised manuscript includes substantially more details on measurements and calculation methods in order for it to be able to stand on its own.

Section 2.1, the site and measurement setup, is included this time. In line 80, we have included details of CRDS measurements at different levels on the tower and EC flux measurements. In the Method section, a new subsection is included to describe the details of calculating Kc. In the section on the method of IDMs, more details were also included.

Line 117. "For these trace gases at this site, the detection limits of this open path system were insufficient . . .". This is very interesting.

Response: We have revised this to "Given the mixture of interfering gas signatures at this site, the detection limits of this open path system were insufficient …"

Line 122. Does the flux calculation really fit the usual description of a modified bowen ratio (MBR) method? It's difficult to know because of uncertainty as to how the fluxes are calculated. Usually an MBR measurement means the unknown flux of the gas of interest is related to the known flux of another tracer gas, plus the gradient of that tracer gas, and the gradient of the gas of interest . No theoretical flux gradient relationships are needed. In this study, it is likely that the tracer diffusivity Kc is estimated from a stability corrected Km measured with a sonic anemometer, I would say the technique is better described as an aerodynamic flux-gradient approach.

Response: We have inserted more details on how the Kc and gradient fluxes were calculated. By using the EC flux of $CH_4$ at 18m, and $CH_4$ gradient measurements on the tower between 8m and 32m, we calculated Kc for $CH_4$. Since the directly calculated Kc time series had many gaps, we took the approach of relating Kc to the more continuously available Km, in order to use the resulting Sc(t) to establish a continuous Kc' time series through Kc' = Km/Sc. This Kc' was then applied to the $NH_3$ and total alkanes gradients. Therefore, our method is closer to MBR than to an aerodynamic flux-gradient approach, although it is true that it is not MBR in the strictest sense. We have replaced all references to the MBR with a more generic "gradient flux method" or similar phrasing.

Line 160: ". . . the warm pond surface . . . resulting in continuing transport of pollutants . . . without significant diurnal variation." When the pond is warm relative to the air, does this mean there are no diurnal changes in atmospheric stability over the pond? I doubt it. This statement also implies that aerodynamic resistance is the limiting factor controlling pond emissions.

Response: Half-hour periods surface turbulence statistics in this study show that the surface layer was unstable (z/L < - 0.0625) 98% of the time when the wind was from the pond. In addition, our measurements in this study only show results during that 5-week period in the summer. Emissions in other seasons remain an unresolved question.

Line 314: "In our modified Bowen ratio approach, Kc is derived from a measured and stability corrected Km . . .". This sentence highlights the confusing description of the methods used in this paper: I do not think Km has been defined to this point in this manuscript; and Eq. (2) suggests that Kc for the FTIR is calculated from Kc taken from point measurements (and not from Km).

Response: We thank the reviewer for pointing this out. As what mentioned in the previous response, we have inserted more details on the methods of calculation fluxes in Section 2.3. Km is now explained. Yes, Kc for the FTIR measurement is calculated from Kc for $CH_4$ from the gradient measurements on the tower.

References:

You, Y., Staebler, R. M., Moussa, S. G., Beck, J., and Mittermeier, R. L.: Methane emissions from an oil sands tailings pond: A quantitative comparison of fluxes derived by different methods, Atmos. Meas. Tech. Discuss., https://doi.org/10.5194/amt-2020-116, in review, 2020.

You, Y., Staebler, R. M., Moussa, S. G., Su, Y., Munoz, T., Stroud, C., Zhang, J., and Moran, M. D.: Long-path measurements of pollutants and micrometeorology over Highway 401 in Toronto, Atmos. Chem. Phys., 17, 14119–14143, https://doi.org/10.5194/acp-17-14119-2017, 2017.

Flesch, T. K., Prueger, J. H., and Hatfield, J. L.: Turbulent Schmidt number from a tracer experiment, Agric. For. Meterol., 111, 299-307, https://doi.org/10.1016/S0168-1923(02)00025-4, 2002.

Gualtieri, C., Angeloudis, A., Bombardelli, F., Jha, S., and Stoesser, T.: On the Values for the Turbulent Schmidt Number in Environmental Flows, Fluids, 2, http://doi.org/10.3390/fluids2020017, 2017.

---

## Author Comment (AC2) · 30 Oct 2020

**Response to Referee #2, "Quantifying fugitive gas emissions from an oil sands tailings pond with open-path FTIR measurements"**

**We thank the referee for the comments and questions. In response, we have added substantial material in order for the manuscript to stand on its own. Below, we address each specific comment in turn. Questions and suggestions are in black, and our responses are in blue.**

For this reason, I recommend that the authors undertake a complete rewrite, with the aim of producing a stand-alone, coherent paper.

Response: We thank the reviewer for the suggestion. We have added substantial material to sections 2.1, 2.2, and 2.3, and have moved Figure S1 in the supplement to the main text, to ensure that this paper can stand on its own in a coherent manner. We have carefully considered all the comments and implemented many of the changes.

The open-path FTIR is subject to density effects due to vertical temperature and humidity gradients. Because they are stronger for gases of lower concentration, ratio-ing the uncorrected molar concentration gradients will not eliminate these effects. I am not sure that I trust their flux values without correcting for these effects.

Response: The temperature measured at 8m was used in retrieving mole fractions for the three paths. The temperature difference shown in You et al. (2020), Figure 4c, is the difference between the pond liquid surface and the temperature measured at 8m, which was usually bigger than the difference between the temperature at 8m and the actual mean temperature for each path. To address this question, we have compared the temperature at 8m to temperature at 1m, and temperature at 18m to temperature at 8m. The statistics are shown here:

[Figure]

**Figure: Histogram of (a) T_8m – T_1m (K) and (b) T_18m – T_8m (K) over the entire study period.**

When the wind was from the pond, (T_8m – T_1m) was mainly between – 1.5 and 0.5 K, and (T_18m – T_8m) was mainly between -0.25 and 0.25 K.

To improve on the method we employed in our original manuscript, we can use T_1m for the bottom path, and an interpolation of T_8m and T_18m at 12m for the top path, to calculate mole fractions. First, the difference between the T_8m (which was used in previous retrieval for all the paths) and T_1m, as well as the difference between T_8m and T_12m, was calculated. In You et al. (2017), the sensitivity of the input temperature on the retrieved mole fractions of $CH_4$ and $NH_3$ was investigated using the same

software (OPUS_RS). Over a 45-degree range (from 278 to 323K), the retrieved mole fraction time series changed by less than 4.2% for $CH_4$ and by less than 8.9% for $NH_3$. We assume that the sensitivity of temperature on retrieved mole fractions of alkanes is similar to $CH_4$, due to the similar absorption mechanisms. The retrieved mole fractions using T_8m (x_original) were corrected as follows:

Bottom path:  x_corrected = x_original + x_original × (T_1m – T_8m)/45 × 4.2%

Top path:  x_corrected = x_original + x_original × (T_12m – T_8m)/45 × 4.2%

For $NH_3$, the 4.2% is replaced by 8.9%.

Next, the $H_2O$ mole fraction was used to calculate dry mole fractions. The temperature and relative humidity at 1m, 8m and 18m were used to calculate $H_2O$ mole fractions at these heights, respectively. The calculated $H_2O$ mole fraction at 1m was used to calculate dry mole faction of gases for the bottom path. The average of the calculated $H_2O$ mole fractions at 8m and 18m was used to calculate dry mole fraction of gases for the top path (with an average height of 12m). The calculated dry mole fraction of $CH_4$ for the bottom path was calibrated with the CRDS $CH_4$ dry mole fraction at 4m, with the exactly same process as illustrated in Figure S1. The calculated mole fractions for top and bottom paths were applied to temperature correction mole fractions to get dry mole fractions. The calculated dry mole fraction of $CH_4$ for the bottom path was calibrated with the CRDS $CH_4$ dry mole fraction at 4m, with the exactly same process as illustrated in Figure S1. Finally, the same gradient flux calculations were performed (using the original Kc-FTIR) with these new temperature corrected dry mole fractions for $CH_4$, $NH_3$ and total alkanes. Results are shown in the Table 1 below. The relative changes in the final fluxes are 8%, 0%, and -25%, for $CH_4$, $NH_3$ and total alkanes respectively. While these changes do not affect the main results and conclusions in this paper, they are significant enough that we included them in our flux calculations in the revised manuscript. Note that the numbers in Table 1 below are different from those presented in the revised manuscript, since the latter incorporate other modifications in response to question 3) of Referee#1.

Table 1: Statistics of gradient fluxes from FTIR with different processes.

| | Flux (g m$^{-2}$ d$^{-1}$) | q_25% | median | q_75% | mean |
|---|---|---|---|---|---|
| CH4 | original | 1.9 | 3.4 | 5.5 | 3.7 |
| | Dry correction | 2.1 | 3.3 | 5.6 | 3.9 |
| | T correction + Dry correction | 2.1 | 3.3 | 5.6 | 4.0 |
| NH3 | original | 0.01 | 0.04 | 0.08 | 0.05 |
| | Dry correction | 0.02 | 0.04 | 0.08 | 0.05 |
| | T correction + Dry correction | 0.02 | 0.04 | 0.08 | 0.05 |
| Total alkanes | original | 0.25 | 0.95 | 1.97 | 1.33 |
| | Dry correction | 0.23 | 0.66 | 1.55 | 1.00 |
| | T correction + Dry correction | 0.23 | 0.67 | 1.55 | 1.00 |

You seem to rely on eddy-covariance methane flux and concurrent measurement of methane concentration gradient to obtain fluxes of other trace gases from the modified Bowen ratio method. How did you get the gradient CH4 flux then? The gradient CH4 flux was biased low in comparison with the eddy-covariance flux. Were other fluxes similarly biased (due to a limited fetch). Can you estimate the "true" emission fluxes of the tailings pond via footprint modeling?

Response: Although no single method exists that can provide a "true" emission flux, eddy covariance is considered the most direct measure of turbulent fluxes between the atmosphere and an underlying surface, since it requires no parameterizations of exchange coefficients / eddy diffusivities, transport modeling or other empirical relationships. Regarding footprint modeling, the Inverse Dispersion Model applied is essentially a footprint model, in that it starts with a defined surface source area and then calculates the emission rate from this source based on a parameterization of the transport mechanisms and the resulting concentration increase observed downwind.

The section on methanol CH3OH should be enhanced. What was the average flux? Did the flux vary with environmental conditions?

Response: we did not discuss $CH_3OH$ much because its mole fraction showed no increase when the wind was from the pond compared to other directions, as shown in Figure S16 (was Figure S17). We attempted to calculate gradient flux of $CH_3OH$ in the same way as for the other pollutants, and the flux was on the order of 1 mg m$^{-2}$day$^{-1}$, but with an uncertainty that made it not statistically different from zero.

The section on comparison with published fluxes is a bit superficial. The reader is interested in knowing if your emission numbers are representative of a typical tailings pond. Also a solid comparison will require footprint correction to your gradient fluxes.

Response: Unfortunately, there is very little published research on quantifying emissions from an oil sands tailings pond. For instance, this study is the first to our knowledge on $NH_3$ emissions. In addition, there is no such thing as a typical tailings pond, in that they serve a large range of purposes, with different chemical and physical characteristics. Tailings ponds also change over the time; previous research has shown that as ponds age, methane emission can start increasing at some point due to the change of microbial activities in the tailings pond (cf. Small et al., 2015). Regarding the footprint, as long as the footprint of the measurement falls within the boundaries of the pond, no correction is required (see Figure 1 (was Figure S1)).

References:

You, Y., Staebler, R. M., Moussa, S. G., Su, Y., Munoz, T., Stroud, C., Zhang, J., and Moran, M. D.: Long-path measurements of pollutants and micrometeorology over Highway 401 in Toronto, Atmos. Chem. Phys., 17, 14119-14143, https://doi.org/10.5194/acp-17-14119-2017, 2017.

Small, C. C., Cho, S., Hashisho, Z., and Ulrich, A. C.: Emissions from oil sands tailings ponds: Review of tailings pond parameters and emission estimates, Journal of Petroleum Science and Engineering, 127, 490-501, https://doi.org/10.1016/j.petrol.2014.11.020, 2015.

---

## Author Response (AR2)

Dear Editorial Team,

During the review process of the associated paper on the same study we currently have in AMTD (amt-2020-116), we discovered that a correction (increase) of the eddy diffusivity by 25% was required due to a computational error. Due to this correction, all the gradient fluxes presented in this current paper (amt-2020-257) also must be increased by 25%. The IDM fluxes and other results remain unchanged, and the correction does not affect any of the conclusions in this paper.

Because the gradient fluxes changed, figures and tables containing gradient flux results have been changed. Affected tables are Table 2 and Table S1. Affected figures are Figure 2, Figure 4, and Figure 5 in the main text, and Figure S7, S8, S9, S11(a), S13, and S14 in the supplement. The marked-up manuscript is inserted below for your reference.

The 25% increase in gradient fluxes does affect the comparison with the IDM flux; the relative difference changed from 11% to 30%. This change is reflected in the second last sentence of the abstract, the last sentence of Section 3.2, and the middle of the first paragraph in Section 4. This does not change our conclusions.

We have presented results of $NH_3$ and total alkanes fluxes with gradient and IDM methods in Table 2. In the original manuscript, we showed flux results from IDM method in the abstract, Section 3.6, and Section 4. When revising the manuscript, we thought showing fluxes from the gradient method in those text is more consistent, since we discussed a limitation of NH3 IDM flux in section 3.3. The agreement of total alkane flux with gradient and IDM methods is also discussed, this time at the end of Section 3.4.

Best regards,
Yuan You & Ralf Staebler
Toronto, 18 December 2020

[revised manuscript text omitted]

**1. Methane mole fractions, vertical profiles, and gradient fluxes**

**Calibration of retrieved $CH_4$ mole fraction from OP-FTIR**

805     The amplitude of spectra for all the three paths varied substantially over the study period, especially for the top path. As a proxy for the spectral amplitude, the signal-to-noise ratio (SNR) of the $CH_4$ fitting was used. The $CH_4$, $NH_3$, $CH_3OH$ and HCHO mole fraction for all three paths when this SNR dropped fast, or stayed below 10 were flagged. 3% and 13% of the measurements from bottom and top path were flagged and invalidated from further mole fraction gradient and flux calculations.

810     Since $CH_4$ mole fraction was also continuously measured by cavity ring-down spectroscopy (CRDS) at four heights during the study, the measurements at 4m were compared to $CH_4$ mole fraction retrieved from the FTIR bottom path to calibrate the retrieved $CH_4$ mole fraction from three paths of this OP-FTIR system.

Each CRDS in this study was calibrated before and after the campaign, and $CH_4$ mole fraction from three CRDS at the same height was well compared ($r^2$>0.96, slope=0.98- 1.01, intercept=0.01-0.02 ppm). Therefore, $CH_4$ mole

815     fraction retrieved from FTIR all three paths were calibrated by the linear relationship in Fig. S2:

$$[CH_4]\_FTIR\_calibrated=1.2015\times[CH_4]\_FTIR\_retrieved - 0.397 \qquad (S.\ 1)$$

[Figure]

**Figure S 1 CH₄ mole fraction retrieved from FTIR bottom path compared to CH₄ mole fraction measured by CRDS (G2204) at 4m. Data are half-hour averaged results.**

820

**Mole fractions and vertical profiles with gradient fluxes**

[Figure]

825     **Figure S 2 Normalised rose plot of CH₄ mole fractions from FTIR bottom path. Colors represent CH₄ mole fractions. The length of each colored segment presents the time fractions of that mixing ratio in each direction bin. The radius of the black open sectors indicates the frequency of wind in each direction bin; angle represents wind direction: straight up is north and straight left is west.**

[Figure]

830

    **Figure S 3 Time series of wind direction, wind speed, difference in CH₄ mole fractions from the top and bottom paths, CH₄ mole fractions, difference in NH₃ mole fractions from the top and bottom paths, and NH₃ mole fractions, from Aug 6th to 8th, and from Aug 27th to Sept 5th. MDT = Mountain Daylight savings Time.**

835    In the analysis of methane vertical profile below, all the mole fractions measurements (half-hour averages) were
       taken from the Picarro G2204 at 4, 8, 18, and 32m. There are 271 half-hours in total when the wind was from the
       pond. About 83% of the half-hour periods when the wind was from the pond direction, the $CH_4$ vertical profiles are
       similar to Fig. S4. Within this 83% of periods, some profiles are close to linear, and others are not strict decreasing
       trend with height. For the rest of 17% of half-hour periods, the $CH_4$ vertical profiles are closer to logarithmic (Fig.
840    S6). Therefore, $CH_4$ vertical profiles are considered linear over the entire period for calculating gradient flux with
       OP-FTIR measurement.
       In addition, those half-hour periods when logarithmic relationship is better than linear to describe the vertical profile
       are mainly (65%) associated with wind speed greater than 6m/s (Fig. S7). For the majority of the time (85%) when
       the wind was from the pond, wind speed was less than 6m/s (Fig. S7).

845

[Figure]

850    **Figure S 4 Examples of observed $CH_4$ mole fractions vertical profile, when the profiles are close to linear.**

[Figure]

**Figure S 5 Examples of observed CH₄ mole fractions vertical profile, when the profiles are close to logarithmic.**

[Figure]

855

**Figure S 6 Time series of wind direction and wind speed measured at 18m over the entire project.**

To compare to the assumption of linear vertical profile of $CH_4$ mole fractions, the calculation of $K_c$ for the assumption of logarithmic vertical profile is also listed here. The representative average height of the FTIR top path

860 with a logarithmic vertical profile would be $Z_{top} = \sqrt{23 \times 1} = 4.8\ m$. Then, $K_c$ for gradient flux calculated from the top-to-bottom path gradient is adjusted logarithmically based on the $K_{c\_2,4}$ calculated from point measurements at 8m and 32m on the tower:

$$F_{gradient_{FTIR}} = -K_{c\_FTIR\_log} \times \frac{\partial c}{\partial z} = \frac{-K_{c_{FTIR\_log}} * \partial c}{z * \ln(\frac{z_2}{z_1})} = -\frac{K_{mFTIR\_log}}{K_{m8,32m}} \times K_{c_{8,32m}} \times \frac{\partial c}{\sqrt{4.79 \times 1} \times \ln(\frac{4.79}{1})} = -0.291 \times$$

$$\frac{K_{mFTIR\_log}}{K_{m8,32m}} \times K_{c_{8,32m}} \times \partial c\ \text{(S. 2)},$$

865 where z is the height for which flux is calculated (Thompson and Pinker, 1981).

$\frac{K_{mFTIR\_log}}{K_{m8,32m}}$ is a function of stability (z/L) and is calculated with eq.(5) and (6) in the main text. The gradient flux of $CH_4$ with logarithmic vertical profile is calculated with eq. (S2) and the area-weighted average flux from the pond sectors is 4.1 $gm^{-2}d^{-1}$, which is 19% greater than the gradient flux calculated with linear vertical profile.

Beside top-bottom paths of $CH_4$ mole fractions gradient, middle-bottom paths of gradient can also be used to

870 calculate $CH_4$ gradient fluxes. The results are summarised in the first row of Table S1 to compare to gradient fluxes with top-bottom paths $CH_4$ gradients. The area-weighted averaged fluxes with middle-bottom paths is 29% lower than the area-weighted averaged fluxes with top-bottom paths (Table S1).

875

[Figure]

**Figure S 7 CH₄ gradient flux from FTIR compared with EC flux.**

880

[Figure]

885 **Figure S 8 Diurnal variation of CH₄ gradient flux from FTIR, when the wind came from the pond direction. MDT = Mountain Daylight savings Time. Lower and upper bounds of the box plot are 25th and 75th percentile; the line in the box marks the median and the black square labels the mean; the whiskers label the 10th and 90th percentile.**

[Figure]

890

**Figure S 9  $CH_4$ gradient flux when the wind was from the pond.**

**IDM flux of CH4 with two approaches of determining background mole fraction input**

IDM fluxes of CH$_4$ with input from FTIR. Fluxes comparison with background mole fraction using ECCC
895    measurement at south, and AEP measurements at north:

**Figure S 10 comparison of CH$_4$ IDM fluxes with input background mole fraction from the south and north measurements.**

900    The half-hour IDM fluxes with these two approaches agree well (slope = 0.9, r$^2$=0.92). The sector-area-weight-
averaged IDM fluxes with two approaches are also within 20% difference. The interquartile ranges overlap (Table
S1).

**2.  NH₃**

905

[Figure]

Figure S 11 Diurnal variations of NH₃ gradient flux derived from top-bottom paths (a) and IDM flux (b) when the wind
910 was from the pond direction. MDT = Mountain Daylight savings Time. Lower and upper bounds of the box plot are 25th
and 75th percentile; the line in the box marks the median and the black square labels the mean; the whiskers label the 10th
and 90th percentile.

915

[Figure]

**Figure S 12 NH$_3$ mole fraction vertical profile after averaging in 16 wind direction sectors. The height z for the three paths are the height of the middle point of each path.**

3. **Total alkane**

[Figure]

**Figure S 13 Total alkane gradient flux compared to CH₄ gradient flux, both derived from OP-FTIR top and bottom paths.**

925

[Figure]

Figure S 14 Diurnal variation of total alkane gradient flux when the wind was from the pond direction. MDT = Mountain Daylight savings Time. Lower and upper bounds of the box plot are 25th and 75th percentile; the line in the box marks the median and the black square labels the mean; the whiskers label the 10th and 90th percentile.

930

[Figure]

935 Figure S 15 Total alkane mole fraction vertical profile after averaging in 16 wind direction sectors. The height z for the three paths are the height of the middle point of each path.

**4. Methanol (CH₃OH)**

[Figure]

 **Figure S 16 CH₃OH mole fraction retrieved from the FTIR bottom path, binned in 22.5° sectors. Lower and upper bounds of the box plot are 25th and 75th percentile; the line in the box marks the median and the black square labels the mean; the whiskers label the 10th and 90th percentile.**

 ### 5. Flux results with the slant path approach from Flesch et al. (2016)

As briefly discussed in the introduction of the main text, Flesch et al., (2016) deployed OP-FTIR measurement with "slant path" configuration, and derived emission rates of $N_2O$ and $NH_3$ by flux-gradient method. To compare the methods we used to calculate gradient fluxes with their approach, we also performed similar calculation. The derived $u_*$ and L directly from sonic anemometer measurement at 8m on the tower, mole fraction difference between

 top and bottom path of FTIR, and calculated $S_c$ were plugged in equation (9) in Flesch et al., (2016). In this study, calculated $S_c$ is allowed to vary with dynamic stability (You et al. (2020) Fig. 3), while in Flesch et al. (2016) $S_c$ was a constant 0.64. The time series of half-hour gradient fluxes of $CH_4$, $NH_3$ and total alkane were calculated. Area weight-averaged fluxes were calculated and summarized in Table S1. Compared to gradient flux results with our approach modified Bowen ratio, $CH_4$, $NH_3$ and total alkane fluxes with the "slant path" flux-gradient method are

 24%, 25%, and 30% smaller.

**Tables**

**Table S1 Summary of CH$_4$ IDM fluxes with two background approaches, and gradient fluxes with approach from Flesch et al. (2016).**

| (g m$^{-2}$ d$^{-1}$) | Q_25% | median | Q_75% | mean[a] |
|---|---|---|---|---|
| CH$_4$_gradient flux with middle-bottom paths | 1.5 | 2.6 | 4.1 | 3.0 ± 1.3 |
| CH$_4$_IDM flux_with ECCC background | 3.6 | 5.2 | 6.6 | 5.4 ± 0.4 |
| CH$_4$_IDM flux_with AEP background | 2.9 | 4.4 | 5.6 | 4.3 ± 0.6 |
| CH$_4$ gradient flux with approach from Flesch et al. (2016) | 1.5 | 2.9 | 4.6 | 3.3 ± 1.3 |
| NH$_3$ gradient flux with approach from Flesch et al. (2016) | 0.01 | 0.03 | 0.06 | 0.04 ± 0.01 |
| Total alkane gradient flux with approach from Flesch et al. (2016) | 0.16 | 0.50 | 1.08 | 0.74 ± 0.15 |

[a] Errors with the mean fluxes are calculated with an integrative approach: the average of observed standard deviations of fluxes from five periods when the fluxes displayed high stationarity.